# Maximizing and evaluating the impact of test-trace-isolate programs: A modeling study

Kyra H. Grantz[1]ᵒ, Elizabeth C. Lee[1]ᵒ, Lucy D'Agostino McGowan[2], Kyu Han Lee[1], C. Jessica E. Metcalf[3,4], Emily S. Gurley[1], Justin Lessler[1]*

1 Department of Epidemiology, Johns Hopkins Bloomberg School of Public Health, Baltimore, Maryland, United States of America, 2 Department of Mathematics and Statistics, Wake Forest University, Winston-Salem, North Carolina, United States of America, 3 Department of Ecology and Evolutionary Biology, Princeton University, Princeton, New Jersey, United States of America, 4 Princeton School of Public and International Affairs, Princeton University, Princeton, New Jersey, United States of America

ᵒ These authors contributed equally to this work.

* justin@jhu.edu

**Data Availability Statement:** All relevant data are within the manuscript and its Supplementary Information files.

**Funding:** This work was supported by funding from State of California Institute of Technology (19-13081) (KHG, ECL, JL), Johns Hopkins

## Abstract

### Background

Test-trace-isolate programs are an essential part of coronavirus disease 2019 (COVID-19) control that offer a more targeted approach than many other nonpharmaceutical interventions. Effective use of such programs requires methods to estimate their current and anticipated impact.

### Methods and findings

We present a mathematical modeling framework to evaluate the expected reductions in the reproductive number, $R$, from test-trace-isolate programs. This framework is implemented in a publicly available R package and an online application. We evaluated the effects of completeness in case detection and contact tracing and speed of isolation and quarantine using parameters consistent with COVID-19 transmission ($R_0$: 2.5, generation time: 6.5 days). We show that $R$ is most sensitive to changes in the proportion of cases detected in almost all scenarios, and other metrics have a reduced impact when case detection levels are low (<30%). Although test-trace-isolate programs can contribute substantially to reducing $R$, exceptional performance across all metrics is needed to bring $R$ below one through test-trace-isolate alone, highlighting the need for comprehensive control strategies. Results from this model also indicate that metrics used to evaluate performance of test-trace-isolate, such as the proportion of identified infections among traced contacts, may be misleading. While estimates of the impact of test-trace-isolate are sensitive to assumptions about COVID-19 natural history and adherence to isolation and quarantine, our qualitative findings are robust across numerous sensitivity analyses.

### Conclusions

Effective test-trace-isolate programs first need to be strong in the "test" component, as case detection underlies all other program activities. Even moderately effective test-trace-isolate

Hospital (ECL, JL), and Bloomberg Philanthropies (KHG, ECL, LDM, KHL, ESG, JL). The funders had no role in study design, data collection and analysis, decision to publish, or preparation of the manuscript.

**Competing interests:** I have read the journal's policy and the authors of this manuscript have the following competing interests: JL is a paid statistical advisor for PLOS Medicine.

**Abbreviations:** ConTESSA, Contact Tracing Evaluation and Strategic Support Application; COVID-19, coronavirus disease 2019; SARS-CoV-2, severe acute respiratory syndrome coronavirus 2.

programs are an important tool for controlling the COVID-19 pandemic and can alleviate the need for more restrictive social distancing measures.

## Author summary

### Why was this study done?

- Control measures for the ongoing COVID-19 pandemic rely largely on untargeted interventions, like social distancing, which have high economic and social costs.

- Test-trace-isolate programs, in which known cases are asked to isolate and their contacts are traced and then asked to quarantine, are an attractive option to control the spread of COVID-19 in a more targeted fashion.

- Estimating the impact of test-trace-isolate programs is not straightforward, due to feed-back loops between control measures and disease transmission.

### What did the researchers do and find?

- We developed a mathematical modeling framework to assess the potential impact of test-trace-isolate programs.

- In most cases, increasing the percentage of cases successfully isolated will yield the larg-est relative reductions in disease transmission (as compared to improvements in suc-cessful contact quarantine or reductions in time to isolation or quarantine).

- Programs already achieving a high percentage of case isolation will see more substantial gains from improving the speed of case isolation and improving the completeness and speed of contact tracing and quarantine.

### What do these findings mean?

- Test-trace-isolate programs must be supported by widespread and expeditious testing and case detection to suppress SARS-CoV-2 transmission.

- Even imperfect test-trace-isolate programs can meaningfully complement other inter-ventions as part of a comprehensive public health response with fewer social and eco-nomic costs.

- The modeling framework is publicly available in an interactive web application, and additional teaching materials are available in a free online course.

## Introduction

In the absence of a vaccine or a widely available prophylactic drug, nonpharmaceutical inter-ventions are the only tools available to curb the spread of the ongoing coronavirus disease 2019 (COVID-19) pandemic. During the initial phases of the pandemic, broad-reaching

interventions like stay-at-home orders and nonessential business closures were applied throughout the world, affecting large swathes of the population and causing severe social and economic disruption [1–6].

Because of the high costs of broadscale social distancing measures, there has been an increasing focus on alternative approaches with fewer ancillary costs. One such approach is a test-trace-isolate program, in which extensive "testing" is used to identify cases in the community; public health agencies then "trace" the contacts of these cases in order to identify people who may have been infected; and the initial cases are asked to "isolate" and their contacts are asked to quarantine for the period of time that they could be, or become, infectious [7]. If effective, test-trace-isolate programs can reduce the need for more restrictive, widespread control measures. Already they have played a critical role in controlling severe acute respiratory syndrome coronavirus 2 (SARS-CoV-2) transmission in places ranging from Utah to South Korea, where these programs have been credited with enabling successful control while avoiding the most restrictive social distancing measures (e.g., stay-at-home orders) [8–11].

Not all test-trace-isolate programs are created equal, and the success of test-trace-isolate programs is ultimately measured in their ability to reduce transmission. The proportion of infections identified through testing and contact tracing dictate the proportion of transmission chains we can potentially disrupt, while the speed of isolation and quarantine dictates how many potentially infectious contacts are prevented [12]. The resulting reductions in disease transmission can be quantified and compared by estimating the reproductive number, $R$, the average number of new infections caused by a single infected individual.

Translating test-trace-isolate program metrics into reductions in the reproductive number is not a direct calculation due to feedback loops between control measures and disease transmission dynamics. As a test-trace-isolate program interrupts chains of transmission, it changes the propagation of disease in future generations. Any approach aiming to translate quantitative program metrics into meaningful measures of the effectiveness of disease control must account for these dynamic processes.

Here, we propose a mathematical framework for modeling the impact of test-trace-isolate strategies on onward transmission, as measured by expected reductions in an average, population-level reproductive number. Using this approach, we explore the factors which most influence the success of a test-trace-isolate program, their interactions, and how these results may be used in developing strategies for improvement. To enable broad adoption of our approach, the methods presented here are implemented in an R package, **tti** [13], and the web-based Contact Tracing Evaluation and Strategic Support Application (ConTESSA) [14], both of which are freely available online.

## Methods

### Mathematical framework

To estimate the effectiveness of a test-trace-isolate program, we frame our analysis in terms of the effective reproductive number, $R$, defined as the number of onward transmissions an infected individual is expected to make given the current immune status of the population and implemented control measures. We first define the population of infected individuals to be spread across 3 compartments; infections **D**etected through testing and subsequently isolated ($D$), infections among **Q**uarantined contacts of identified cases ($Q$), and undetected infections in the **C**ommunity ($C$) (Fig 1). Though there will be uninfected individuals both in quarantine and in the community, these do not play a role in our calculations and are ignored.

We consider the proportion of the population in each compartment at any given time $t$ to be defined by a 1×3 matrix, denoted $DQC_t$. To calculate the proportion of infected individuals

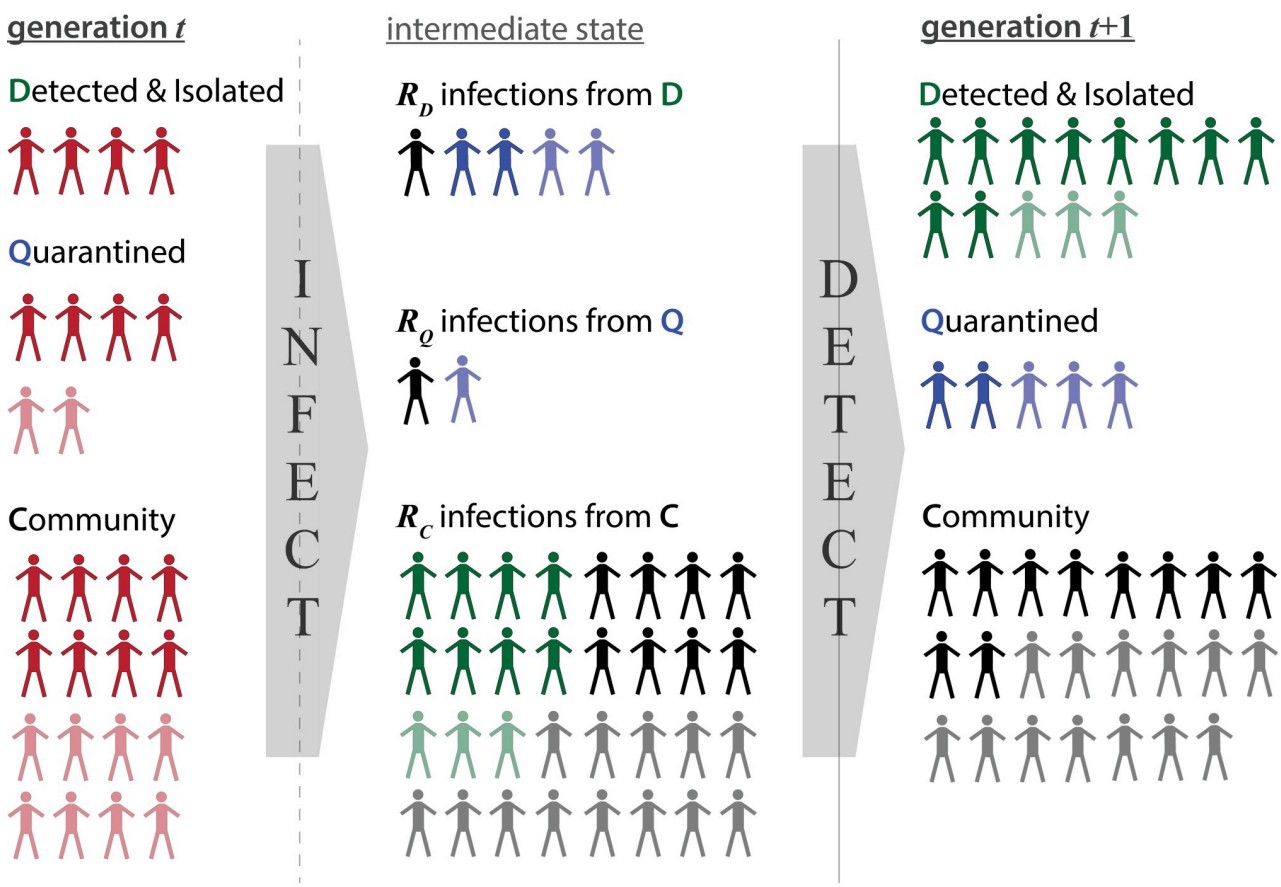

**Fig 1. Conceptual representation of the model algorithm**, where infections in generation $t$ (left column) infect new individuals according to $R_D$, $R_Q$, and $R_C$ reproductive numbers that populate the *INFECT* matrix (center column). These newly propagated infections are then distributed into $D$, $Q$, and $C$ compartments in generation $t+1$ (right column) according to the various detection transition probabilities specified in the *DETECT* matrix (colors of center column). Symptomatic individuals (darker shading) may be more likely to be detected than asymptomatic individuals (lighter shading).

in each compartment at time $t+1$, we apply the rates at which individuals in each compartment cause new infections, and then the rates at which these secondary infections are detected and isolated through the following equation:

$$DQC_{t+1} = \frac{(DQC_t)(INFECT)(DETECT)}{\sum[(DQC_t)(INFECT)]},$$    (1)

where *INFECT* is a 3×3 diagonal matrix describing the number of infections caused in the next generation by members of each detection compartment, and *DETECT* is a 3×3 matrix describing the probability that infections in the next generation are detected and isolated, quarantined, or undetected in the community.

The diagonal elements of the *INFECT* matrix, $[R_D, R_Q, R_C]$, represent the reproductive number for members of each compartment. Hence, given the normalized version of the *DQC* matrix specified above, we can calculate the overall reproductive number at time $t$ as:

$$R_t = (DQC_t)(INFECT)$$

The *DETECT* matrix then assigns these new infections to the appropriate detection classes of the *DQC* matrix in the next generation. Specifically:

$$DETECT = \begin{bmatrix} I(D) \rightarrow D & I(D) \rightarrow Q & I(D) \rightarrow C \\ I(Q) \rightarrow D & I(Q) \rightarrow Q & I(Q) \rightarrow C \\ I(C) \rightarrow D & 0 & I(C) \rightarrow C \end{bmatrix}$$

$$= \begin{bmatrix} (1-\omega_D)\rho & \omega_D & (1-\omega_D)(1-\rho) \\ (1-\omega_Q)\rho & \omega_Q & (1-\omega_Q)(1-\rho) \\ \rho & 0 & (1-\rho) \end{bmatrix} \qquad (2)$$

where *I(X)* represents those infected by people in compartment *X* in the previous generation, $\omega_X$ is the probability that a contact of a detected individual in compartment *X* is traced and quarantined (quarantine completeness), and $\rho$ is the probability that a community infection is detected and effectively isolated by a test-trace-isolate program (isolation completeness). The transitions in each row of the *DETECT* matrix represent the probability that people in the corresponding notional infection compartment will be detected by a particular means (hence rows sum to one).

The other aspect that determines the effectiveness of a test-trace-isolate program is the reduction in *R* that we see among those who are isolated or quarantined (Fig 2). We define the

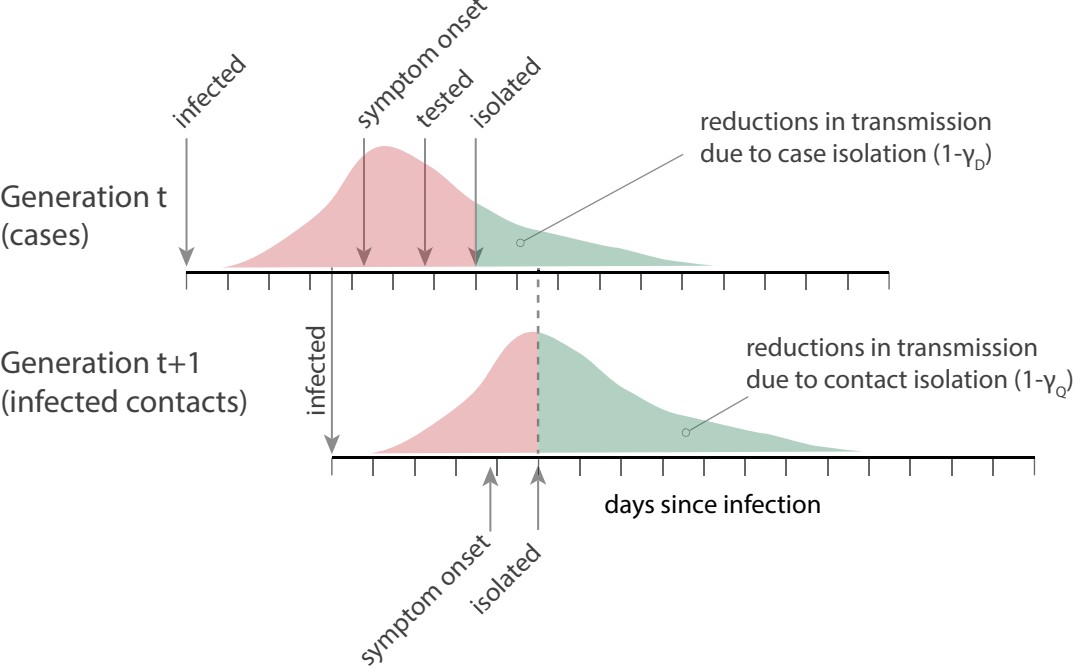

**Fig 2. Conceptual representation of test-trace-isolate programs, where detection of a case in generation *t* through widespread testing and subsequent isolation reduces onward transmission to generation *t+1*.** Individuals in generation *t+1* are then traced and quarantined (and subsequently isolated, if the contact is a suspected or confirmed case) to reduce onward transmission from those who may be infected.

relative transmissibility of individuals in these compartments by $\gamma_D$ and $\gamma_Q$ such that:

$$R_D = \gamma_D R_C \tag{3}$$

$$R_Q = \gamma_Q R_C \tag{4}$$

The main mechanism by which isolation and quarantine reduce transmission is the (at least partial) truncation of the infectious period. Hence, these values are defined by the equations:

$$\gamma_D = \int_{-\infty}^{\tau_D} f(x)dx \tag{5}$$

$$\gamma_Q = \int_{-\infty}^{\tau_Q} g(x)dx \tag{6}$$

where $f(x)$ is the distribution of relative infectiousness indexed from day of symptom onset; $g(x)$ is the expected distribution of relative infectiousness of secondary cases indexed from their infector's time of symptom onset; and $\tau_D$ and $\tau_Q$ represent the average time to case isolation and contact quarantine, respectively, from time of case symptom onset. We therefore assume isolation and quarantine are perfectly effective, such that individuals do not transmit once in quarantine or isolation, though exceptions to this assumption are discussed below.

## Translating observed metrics to model inputs

Health departments collect data on their test-trace-isolate programs, but these observed metrics need to be translated into model inputs. To estimate isolation completeness ($\rho$), we can divide the average number of infections that were isolated by the estimated number of total infections in the community. The latter value may be difficult to obtain but can be approximated in a number of ways, including serosurveys and extrapolation from the number of deaths and approximate infection fatality ratio. Quarantine completeness ($\omega$) can be estimated by dividing the number of quarantined individuals by the total number of contacts.

For the purposes of our model, time to isolation ($\tau_D$) is the average number of days from case symptom onset to case isolation, while time to quarantine ($\tau_Q$) is the average number of days from case symptom onset to contact quarantine. Often these timings are the composite of several constituent processes, including time from symptom onset to testing, time from testing to notification and isolation, and the time from obtaining a test result to tracing and quarantining contacts.

## Adding real-world complexity to the model

Above we describe the basic model framework, but to take into account the complexities of the real world, we can expand the number of compartments in the *DQC* matrix and the corresponding *INFECT* and *DETECT* transitions. In the implementation used to generate the results described below, we create compartments to differentiate symptomatic and asymptomatic infections as well as household and community contacts to address the fact that these groups may differ in their probability and speed of detection, ability to be traced and quarantined, infectiousness, and risk of being infected.

Contact quarantine may not perfectly disrupt onward transmission in the real world; average quarantine duration may be modified in the expanded model to match local guidelines, and imperfect quarantine adherence may be incorporated by tuning the proportion of contacts assumed to be effectively quarantined. The assumption of perfectly effective case isolation may be modified by tuning the proportion of cases assumed to be effectively isolated.

This expanded model has 9 *DQC* compartments and is described in full in the Supplementary Methods (S1 Text).

## Disease simulation

To obtain expected reductions in *R*, we first initiate the model with a *DQC* matrix that has only undetected community infections ($C = 1$). Then, we simulate the infection and detection processes across multiple disease generations until equilibrium is achieved in the *DQC* matrix (this usually occurs in less than 10 generations). This equilibrium provides an estimate of what would be achieved by a specified test-trace-isolate strategy if it maintained its current characteristics for the foreseeable future.

The model may also be run as a stochastic simulation, to explore the impact of overdispersion and stochasticity in the epidemic process in general. In stochastic simulations, we normalize the *DQC* matrix to a standard population size, then simulate the number of infections each individual causes for each notional *INFECT* compartment as a random draw from a negative binomial distribution based on the values in the *INFECT* matrix. These infections are then assigned to the compartments *DQC* matrix based on draws from a multinomial distribution parameterized by the rows of the *DETECT* matrix. Since there is no equilibrium state, stochastic simulations are run for a fixed number of generations.

The simulations presented in this paper assume that $R_0 = 2.5$ without interventions, a generation time of 6.5 days (unless otherwise stated, Table C in S1 Text) [15], and initially, that whether individuals develop symptoms or not has no meaningful impact on transmission or detection probability in surveillance, and that household and community contacts are equally likely to be infected and quarantined. In later analyses, we relax these latter assumptions and add stochasticity to illustrate how our results are influenced by overdispersion in transmission, the presence of asymptomatic transmission, and differential risk of infection and tracing speed in household contacts.

All scenarios are based on hypothetical test-trace-isolate programs designed to represent a range of possible program effectiveness. Other parameter assumptions are available in Table C in S1 Text.

## Model code and resources

The expanded model is implemented in the **tti** R package [13]. We also developed the ConTESSA, an R Shiny web application, around a simplified version of our modeling framework. The purpose of this application is to provide a user-friendly interface where managers of test-trace-isolate activities, equipped with their observed metrics, can examine how well their program reduces onward transmission and explore how their results might change with improvements to completeness and timing metrics as well as different underlying assumptions. The ConTESSA application is complemented by a free Coursera course that describes key contact tracing program metrics and provides more detailed instruction on how to use the application [16].

## Results

### Case isolation and contact quarantine completeness and timing

Application of our framework shows that the reproductive number can be reduced by improving performance across all dimensions of a test-trace-isolate program: detection and isolation completeness, the speed of case isolation, the proportion of contacts followed up and quarantined, and the speed at which quarantine occurs (Fig 3). However, the effect of improvements

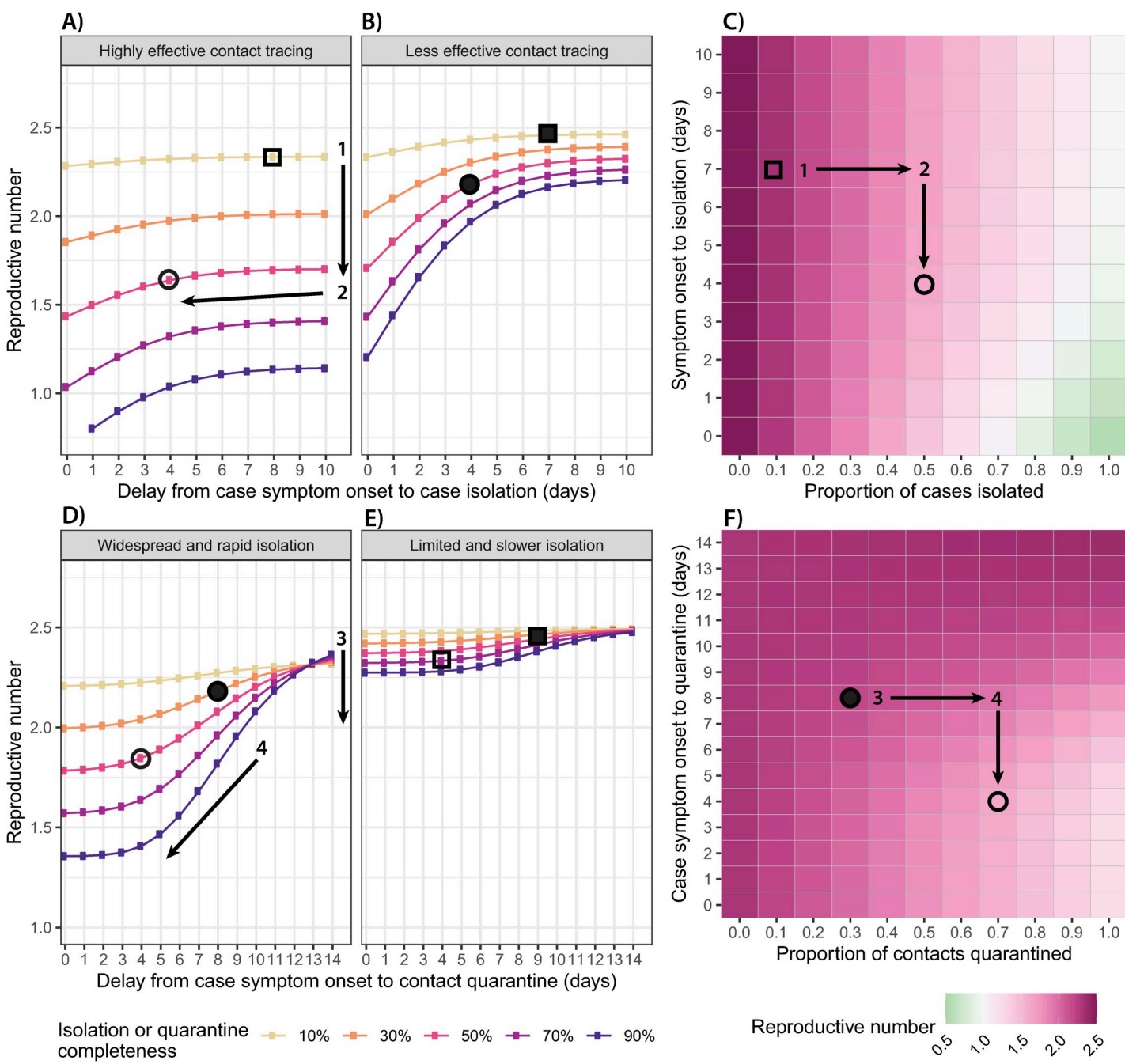

**Fig 3. Improvements to case isolation and contact quarantine**: Impact of case isolation timing (x-axis) and completeness (line colors) on the effective reproductive number (y-axis) for **(A)** a highly effective contact tracing program and **(B)** a less effective contact tracing program. **(C)** Heat map of the effective reproductive number across a range of case isolation timing (y-axis) and completeness (x-axis) scenarios, assuming that contact tracing is highly effective. Impact of contact tracing timing (x-axis) and completeness (line colors) on the effective reproductive number (y-axis) for **(D)** a widespread and rapid case isolation scenario and **(E)** a less effective and slower case isolation scenario. **(F)** Heat map of the effective reproductive number across a range of contact tracing timing (y-axis) and completeness (x-axis) scenarios, assuming that detection and isolation of index cases is widespread and rapid. For all panels, the open shapes mark example scenarios with highly effective contact tracing (70% quarantined on average 4 days after case symptom onset) in contrast to the filled shapes of a less effective contact tracing scenario (30% quarantined after 8 days). Circles mark example scenarios with widespread and rapid case isolation (50% isolated on average 4 days after case symptom onset) in contrast to squares, which have limited and slower case isolation (10% isolated after 7 days). Shapes display consistent scenarios across all panels in the figure.

in one dimension are not independent of performance in the other dimensions, and as such, the aspect where improvements will yield the greatest dividends, the "next best move," depends on the current status of the program.

Consider a situation where you only detect and isolate 10% of cases through your community testing program, and it takes an average of 7 days from symptom onset to do so (squares in Fig 3). Whether you have highly effective contact tracing (70% of contacts quarantined on average 4 days after case symptom onset, hollow shapes in Fig 3) or less effective contact tracing (30% quarantined on average 8 days after case symptom onset, solid shapes in Fig 3), little will be gained by improving the speed of case isolation, and the greatest reductions in transmission will be achieved by improving the proportion of infections detected and isolated (direction 1 in Fig 3A). Increasing the proportion of infections detected and isolated, regardless of other metrics, is critical because it is these detected cases that "seed" all other test-trace-isolate activities. If an infection is not detected, it cannot be isolated, and their contacts cannot be traced and quarantined.

Reducing the time to case isolation (direction 2 in Fig 3A) may be nearly as effective as improving case isolation completeness in certain contexts, in particular, if contact tracing is less effective and isolation completeness is reasonably high (over 30%). Because the vast majority of transmission occurs in the days immediately before and after symptom onset, improvements in the speed of case isolation that bring it to 4 days or fewer will yield the greatest reductions in transmission. Beyond this, there is limited opportunity to reduce onward transmission of the isolated case and thus little difference between delays of 6, 8, or 10 days. Increasing the speed of case isolation is particularly important when contact tracing is ineffective (Fig 3B), hence the program must rely predominantly on reductions in transmission from isolated cases themselves.

The benefits of increasing the speed and completeness of contact tracing and quarantine are greatest in the context of already highly effective testing and isolation (e.g., 50% of infections isolated on average 4 days after symptom onset, circles in Fig 3D). In such situations, increasing quarantine completeness yields the largest reductions in transmission (direction 3 in Fig 3D), particularly when the speed of contact quarantine is less than 8 days from case symptom onset.

Improvements in the speed of contact quarantine (direction 4 in Fig 3D) are most effective during the 4- to 8-day window after case symptom onset for similar reasons; this period corresponds to the greatest expected infectiousness of infected contacts. The impact of improvements in the speed or completeness of contact tracing is less if testing and isolation fail to reach a high percentage of infections (Fig 3E). As noted above, this is because without adequate detected cases to seed contact tracing activities, the ability of tracing and quarantine to have an impact is substantially reduced.

The completeness and timing of test-trace-isolate activities determine whether we can achieve goals in disease control using these approaches. A common goal is to bring $R$ below one so that the epidemic will start to recede in the population. We explored the range of program characteristics that could achieve this goal, under the assumption that contacts were quarantined on the same day as case isolation, and quarantine completeness ranged from 50% to 100% (Fig 4).

At $R = 2.5$ (our baseline when no other interventions are in place) with perfect contact quarantine, at least 60% of cases must be isolated to achieve $R<1$; this minimum percentage decreases to 50% when starting at $R = 2$, 34% at $R = 1.5$, and 20% at $R = 1.25$ (corresponding to 20%, 40%, and 50% reductions in baseline $R$ due to other interventions) (Fig 4). It is not possible to achieve $R<1$ if case isolation occurs more than 6.4 days after symptom onset, on average, and this threshold increases to 7, 9, and 10 days for $R$ starting at 2, 1.5, and 1.25.

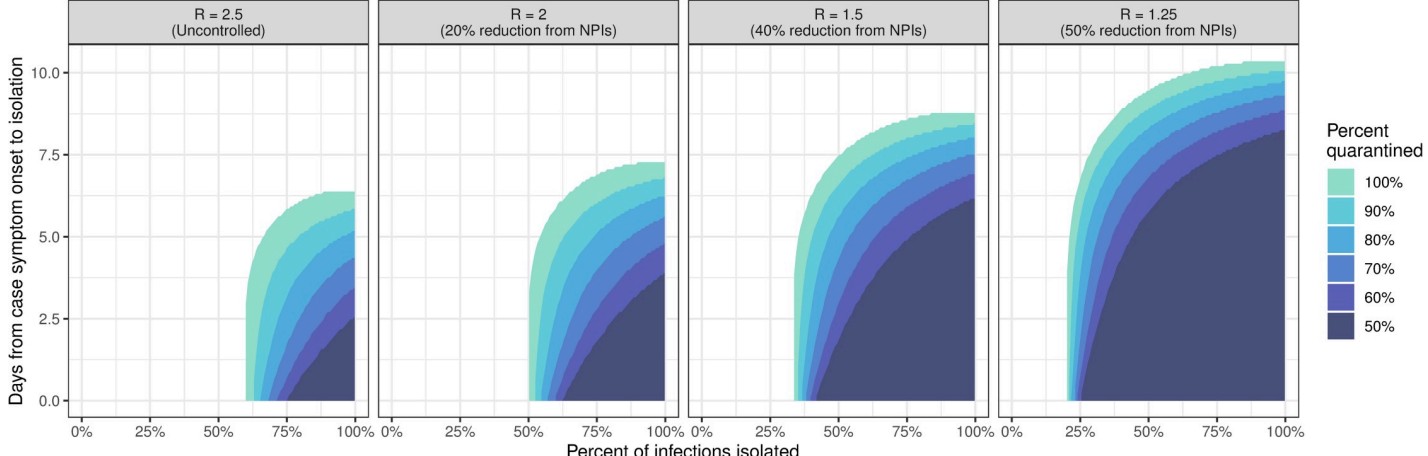

**Fig 4. Isolation strategies (timing and completeness) capable of achieving *R*<1** when a given proportion of contacts (50% to 100%) are quarantined on the same day as case isolation. These strategies are shown for 4 possible baseline values of *R*, assuming other nonpharmaceutical interventions (NPIs) are in effect to reduce transmission from the uncontrolled scenario, *R* = 2.5.

Quarantine need not be perfect to achieve *R*<1 (colors in Fig 4), but the tolerance for imperfect quarantine changes with isolation speed. For example, case isolation completeness needs to increase from 60% to 75% to offset a decrease in contact quarantine completeness from 100% to 50% when cases are isolated on the same day as symptom onset. However, this tolerance for incomplete quarantine rapidly degrades as the time to isolation increases, and, in the absence of other interventions (i.e., *R* = 2.5), no isolation program will achieve *R*<1 with 50% quarantine completeness if the average time to case isolation exceeds 2.6 days.

### Infections arising in traced contacts

A metric often recommended for evaluating test-trace-isolate programs is the proportion of identified infections that were already under quarantine [7]. Programs are thought to be successful if this metric is high because it may indicate that a substantial fraction of transmission chains were interrupted through isolation of case contacts early in (or prior to) their infectious period. In practice, a program might approximate this by calculating the proportion of all newly detected infections among traced contacts, who may or may not be in quarantine when they are confirmed to be infected. This proportion is equivalent to $\frac{Q}{Q+D}$ in our model.

An increasing proportion of new infections detected among traced contacts only represents an improvement in the program effectiveness if the timing of case isolation remains constant or becomes quicker. As delays in isolation increase, a greater number of secondary infections will occur among traced contacts. Hence, we can see both an increase in the proportion of newly detected infections among traced contacts and an increase in the effective reproductive number (direction "2" in Fig 5).

### Adding real-world complexity

In the sections above, we assume that all infected individuals are equally likely to transmit and be detected. Yet, there is evidence that asymptomatic individuals (those who never develop symptoms) are less infectious than those who do develop symptoms [17,18]. If the relative contribution of asymptomatic individuals to onward transmission is low, limited detection of asymptomatic cases will have little effect on the reproductive number (Fig A in S1 Text).

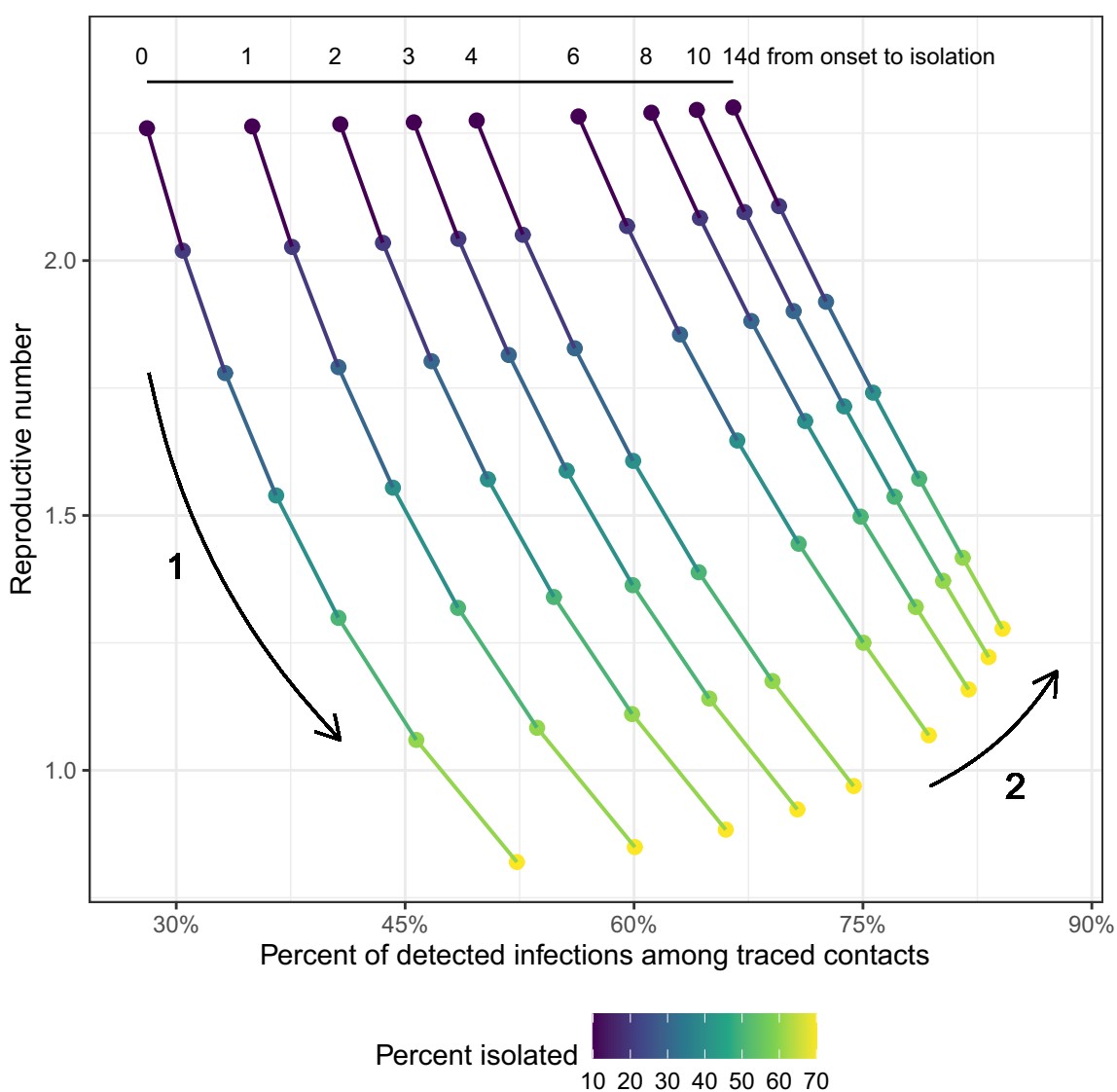

**Fig 5. Relationship between *R* and the proportion of detected infections among traced contacts.** Each position along a line shows a single test-trace-isolate strategy, with a fixed delay from case symptom onset to isolation (shown in the numbers at the top) and 90% of contacts quarantined on the same day as case isolation. Points are colored by the proportion of infections detected and isolated through testing. When isolation timing remains constant, higher case isolation completeness corresponds with increases in the proportion of detected infections among traced contacts and reductions in *R* (Arrow 1). However, increases in the proportion of detected infections among traced contacts could indicate an increase in *R*, if the delay to case isolation is also increasing, thus leading to more secondary cases among traced contacts (Arrow 2).

Similarly, previous scenarios assume that all contacts are equally at risk of being infected, but contact tracing studies suggest that the risk of SARS-CoV-2 infection for household contacts could be 6 (or more) times higher compared to other close contacts [11,17]. Quarantining household contacts therefore is expected to have a larger impact in reducing the reproductive number than quarantining nonhousehold contacts (Fig B in S1 Text).

There is growing evidence that the distribution of onward transmission for SARS-CoV-2 is overdispersed, meaning that most individuals contribute little to onward transmission, while superspreading events result in the bulk of secondary infections [19,20]. Overdispersion has

little effect on the mean estimates of the reproductive number (Fig C in S1 Text). However, when there are few total infections in a community, the random occurrence or detection of any one superspreading event can drastically alter program impact, including ending an outbreak altogether, thereby increasing uncertainty in *R*.

Recent policy recommendations have suggested that 10-day quarantines rather than the previously recommended 14-day quarantines may be acceptable under certain circumstances [21]. When we modify our assumption of perfectly effective quarantine, we find that 10-day quarantines are somewhat less effective and that this effect is more pronounced in scenarios with rapid quarantine and widespread and rapid isolation (Fig D in S1 Text).

### Sensitivity analyses

The above results are sensitive to assumptions about the infectious period and generation time of SARS-CoV-2. While the qualitative trends and relationships hold regardless, assumptions of a shorter generation time substantially increase the necessary speed at which activities must occur to have a meaningful impact (Fig E–I in S1 Text).

## Discussion

Test-trace-isolate programs have the potential to play an important role in COVID-19 control, but the extent of that role depends on each program's ability to limit transmission of the virus. Here, we have presented a modeling framework with which to evaluate the performance of the often independent test-and-isolate and contact tracing components of test-trace-isolate programs and shown how 4 key metrics of performance interact. The results show that a program's ability to detect cases through community testing is one of the greatest drivers of program effectiveness, as it is these cases that seed all other activities. Rapid case isolation, complete contact tracing, and timely quarantine will compound the impact that identifying a new seed will have on the trajectory of the epidemic. Nevertheless, exceptional performance may be needed across all of these dimensions if transmission is to be controlled by test-trace-isolate alone, raising the importance of complementary control activities [22].

Previous work has emphasized the importance of decreasing delays to case isolation and contact quarantine to reduce disease transmission [23,24]. However, these models assumed high testing and isolation coverage (≥60%), levels perhaps attainable in the early or late stages of an outbreak, but far exceeding estimated infection detection rates in many locations with established epidemics [25–28]. Here, we explore a broader range of case detection and isolation proportions, including values that may be more feasibly attained in areas with substantial incidence. In doing so, we show the critical importance of the proportion of cases detected.

Once adequate levels of detection are reached, the speed of case isolation and contact quarantine will begin to matter more. In practice, these delays are linked, as many of the same activities must be completed before isolation or quarantine can occur (e.g., sample collection, laboratory testing, contact of infected or exposed individuals). Hence, practical improvements in important aspects of a test-trace-isolate program, such as speed of laboratory testing, may have an outsized impact on the performance of the program overall.

These results have implications for the design and implementation of test-trace-isolate programs. The ability to identify and interrupt a high proportion of transmission chains will require investment in widespread testing. Population screening, in addition to symptom-based or at-will testing, could meaningfully improve a program's impact on transmission. Inexpensive, easy to administer, and widely available tests could facilitate expanded detection efforts [29,30]. Even if such tests have lower sensitivity, the higher coverage attainable could result in detection of a higher percentage of infections [31]. Importantly, these improvements

to testing capacity and access must be available to those populations most impacted by COVID-19 and not exacerbate existing testing disparities. If we increase testing coverage, but fail to reach those who are at highest risk for infection, there will be little impact on transmission.

Any increases in case detection will be ineffective if isolation and quarantine fail to interrupt transmission. To some extent, this can be counteracted by improvements to other aspects of a program (e.g., proportion of cases detected), but measures to facilitate effective isolation and quarantine may be needed to reach transmission control goals. Given the high risk of household transmission, measures that can intervene in these settings may be highly important. Facilities for isolation outside of the household were associated with substantial reductions in $R$ in China [32]. Use of facial coverings has also been proven effective at reducing household transmission for SARS-CoV-2 and other respiratory viruses [33–35]. Social and economic support for isolation and quarantine, particularly if outside the home, may be necessary for test-trace-isolate programs to be effective in many communities [7]. Technology, such as use of mobile phone data to identify contacts, may improve the speed and completeness of contact tracing, but compliance may be undermined if contact definitions are too broad, and substantial resources would still be required to initiate and support quarantine of identified contacts [36,37].

The case isolation and contact quarantine completeness metrics used in our model may be better conceptualized as the "proportion of onward transmission that would have been otherwise caused by those isolated or quarantined." Thus, programs will see greater reductions in transmission from isolation or quarantine of those who are more likely to transmit. In the expanded version of our model, we already address differences between symptomatic and asymptomatic transmission and household versus community contacts. There may be programmatic reasons to specifically target other groups, and our basic framework could be extended to accommodate such strategies.

Our framework is not without limitations. This model describes a general strategy of tracing and quarantining the immediate contacts of identified cases in a community. It may not be easily extensible to settings such as schools or workplaces. Alternative strategies also exist, including tracing of contacts-of-contacts [38] and so-called backwards tracing [39]. The latter approach is part of a fundamentally different way of using contact tracing in disease control that has been used in COVID-19 response in some countries (e.g., Japan [40]), which focuses on identifying settings that have the potential to facilitate superspreading events or otherwise amplify transmission.

Our estimates of program effectiveness are sensitive to assumptions of disease natural history, though we have explored the impact of some of these assumptions (Fig E–I in S1 Text), and further exploration is possible. We assumed that the incubation period, generation time, and test-trace-isolate time delays were fixed at mean values, although previous work has shown that variability in these time delay distributions may affect the impact of disease control [12]. In addition, the model relies on a simplified version of transmission that does not account for many risk factors for SARS-CoV-2 infection or the contact structure in the population [38,41], which could lead to persistent transmission even when the population reproductive number is low (e.g., if there are clusters of connected cases that are not detected or cannot isolate completely). This may be particularly true if heterogeneities in transmission are associated with our ability to identify individuals through testing or contact tracing. Though nonadherence can be crudely considered by modifying the proportion isolated and proportion quarantined, our model framework does not capture the real-world imperfections where cases and contacts are unable to completely avoid contact with all other individuals. While the reproductive number is a useful representation of transmission control at the population level,

it does not capture differential health burden of infections, and a program could have a higher $R$ but better limit mortality if it effectively protects at-risk populations.

Test-trace-isolate programs should not alone be used to control COVID-19. It is exceedingly difficult to achieve a reproductive number less than one without additional reductions in transmission from other interventions, which may include masking or broadscale social distancing. Likewise, our model is based on proportions, but resource needs are determined by the absolute number of identified cases and traced contacts. When incidence is high, resources needed to test and trace the desired proportion of infections may be excessive, but if incidence is brought down through other control measures, the same targets may be logistically feasible. Still, test-trace-isolate programs need not themselves bring $R$ below one to be valuable, and incremental reductions in transmission can alleviate the need for the most severe social distancing measures.

Hence, test-trace-isolate isolate programs have a valuable role to play in the COVID-19 response, even if such programs themselves are not the whole solution. Understanding the impact that a program is having, or that would result from investing in better program performance, is critical to the effective use of these programs within a broader control strategy. We hope that the model presented here, which is implemented in publicly available tools, will help facilitate the effective use of these programs in the COVID-19 response.

## Supporting information

**S1 Text. Supporting information.** Model details, input parameters, and sensitivity analyses. (PDF)

## Acknowledgments

We would like to thank the numerous public health workers and agencies that gave comments and guidance on the development of our model and the ConTESSA app, including Josh Sharfstein and Melissa Marx. We would also like to thank Isabel Rodríguez-Barraquer for her valuable comments on this manuscript. Finally, we recognize the valuable support from the Johns Hopkins Bloomberg School of Public Health Center for Teaching and Learning.

## Author Contributions

**Conceptualization:** Kyra H. Grantz, Elizabeth C. Lee, C. Jessica E. Metcalf, Justin Lessler.

**Formal analysis:** Kyra H. Grantz, Elizabeth C. Lee, Lucy D'Agostino McGowan, Justin Lessler.

**Funding acquisition:** Kyu Han Lee, Emily S. Gurley, Justin Lessler.

**Methodology:** Kyra H. Grantz, Elizabeth C. Lee, Lucy D'Agostino McGowan, Kyu Han Lee, C. Jessica E. Metcalf, Emily S. Gurley, Justin Lessler.

**Software:** Kyra H. Grantz, Elizabeth C. Lee, Lucy D'Agostino McGowan.

**Visualization:** Kyra H. Grantz, Elizabeth C. Lee, Justin Lessler.

**Writing – original draft:** Kyra H. Grantz, Elizabeth C. Lee, Justin Lessler.

**Writing – review & editing:** Lucy D'Agostino McGowan, Kyu Han Lee, C. Jessica E. Metcalf, Emily S. Gurley.

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
