## [Editor Report · Decision Letter 0]

3 Sep 2020

Dear Dr Lessler, 

Thank you for submitting your manuscript entitled "Maximizing and evaluating the impact of test-trace-isolate programs" for consideration by PLOS Medicine.

Your manuscript has now been evaluated by the PLOS Medicine editorial staff and I am writing to let you know that we would like to send your submission out for external peer review.

Kind regards,

Helen Howard, for Clare Stone PhD 

Acting Editor-in-Chief

PLOS Medicine 

plosmedicine.org

---

## [Decision Letter · Decision Letter 1]

11 Dec 2020

Dear Dr. Lessler,

Thank you very much for submitting your manuscript "Maximizing and evaluating the impact of test-trace-isolate programs" (PMEDICINE-D-20-04296R1) for consideration at PLOS Medicine. 

Your paper was evaluated by a senior editor and discussed among all the editors here. It was also discussed with an academic editor with relevant expertise, and sent to three independent reviewers. The reviews are appended at the bottom of this email and any accompanying reviewer attachments can be seen via the link below:

[LINK]

In light of these reviews, I am afraid that we will not be able to accept the manuscript for publication in the journal in its current form, but we would like to consider a revised version that addresses the reviewers' and editors' comments. Obviously we cannot make any decision about publication until we have seen the revised manuscript and your response, and we plan to seek re-review by one or more of the reviewers. 

We expect to receive your revised manuscript by Jan 01 2021 11:59PM. Please email us (plosmedicine@plos.org) if you have any questions or concerns.

We look forward to receiving your revised manuscript. 

Sincerely,

Emma Veitch, PhD

PLOS Medicine

On behalf of Richard Turner, PhD, Senior Editor, 

PLOS Medicine

plosmedicine.org

*In line with the usual journal style and to support information retrieval in search indexes etc, we'd suggest including some designation of the study design/methodological approach in the article title (normally this is in a subtitle after a colon, and here we'd imagine something along the lines of "....: modelling study" might be appropriate).

*PLOS Medicine style for the abstract normally involves including a summary/note about any key limitations of the study methods in the abstract (normally in the last sentence of the Abstract Methods and Findings section). We noted that the reviewers raise queries about the extent to which compliance or adherence with isolation measures are factored into the model - so perhaps this could be mentioned. (see also below for the main discussion section).

*Because the paper addresses a topic of extreme current importance to public health, and the findings could directly influence policy, we'd imagine it may be (if accepted and published) widely read. Therefore it would be good to make sure the abstract, particularly, is framed in a way that can be more clearly understood by nonspecialists and perhaps even lay readers. Some sentences currently are a bit cryptic for non-specialists - eg the following, and we'd suggest reframing this (and some others) so the meaning is more clear: "**Formally framing the dynamical process** also indicates that metrics used to evaluate performance of test-trace-isolate, such as the proportion of identified infections among traced contacts, may be misleading". (The part in asterisks is the bit that many readers may stumble over).

*At this stage, we ask that you include a short, non-technical Author Summary of your research to make findings accessible to a wide audience that includes both scientists and non-scientists. The Author Summary should immediately follow the Abstract in your revised manuscript. This text is subject to editorial change and should be distinct from the scientific abstract. Please see our author guidelines for more information: https://journals.plos.org/plosmedicine/s/revising-your-manuscript#loc-author-summary

*Please clarify in the paper if the analytical approach followed here was set out prospectively - please state this (either way) early in the Methods section.

c) In either case, changes in the analytical approach -- including those made in response to peer review comments-- should be identified as such in the Methods section of the paper, with rationale.

*As noted above for the abstract - it's not clear that one of the main limitations raised by reviewers - that of adherence to isolation measures - is taken into account in the models. If not, and/or if only done so to a limited extent, we'd suggest including this in the Limitations section as the reviewers note. 

Comments from the reviewers:

Reviewer #1: The authors present a mathematical modeling framework aiming to evaluate the expected reductions in the reproductive number, R, from test-trace-isolate programs.

Comments:

This is a well written, important and timely article.

The mathematical model is presented clearly and concisely. The framework appears to be appropriate for the requirements of this modelling exercise.

Did the authors consider 'adherence or compliance to isolation or quarantine requests' as a parameter of interest?

The model treats all cases as equal, and does not include any element of behavioural effect or community exposure. For instance, a case where someone is working from home without the need to travel and with no dependents nor contacts in the community is different to a case where someone uses public transport on a daily basis to a communal workplace, with children at school, and an active social life. 

This is acknowledged to some extent, but could be expanded upon, within the limitations, which state:

"This model describes a general strategy of tracing and quarantining the immediate contacts of identified cases in a community. It may not be easily extensible to settings such as schools or workplaces. ... The model relies on a simplified version of transmission that does not account for many risk factors for SARS-CoV-2 infection or the contact structure in the population, which could lead to persistent transmission even when the population reproductive number is low." 

Furthermore, the authors acknowledge that "while the reproductive number is a useful representation of transmission control at the population level, it does not capture differential health burden of infections, and a program could have a higher R but better limit mortality if it effectively protects at-risk populations."

These caveats for accurate interpretation and application of the model should also be stated more clearly throughout the manuscript.

Overall, this study provides valuable insight to the elements of test-trace-isolate that have greatest impact on the transmission of SARS-CoV-2 infection, and how these elements relate with each other based on the models' assumptions of the disease natural history. Instrumentally, the model has been made publicly available, ready for further research and development, as well as the exploration of various what-if scenarios.

Reviewer #2: This is a nice piece of work describing a mathematical model of the potential impact of test, trace and isolate (TTI) on the reproduction number (R) of SARS-CoV-2. The work explores the impact of different levels of case detection, quarantine effectiveness and timeliness. It presents interesting findings on the proportion of contacts that are already quarantined, which is often used as a metric of TTI performance and yet depends strongly on the nature of the epidemic (e.g. R) and may in fact be inversely related to some measures of performance such as timeliness. The work is also supported by the provision of an online tool and comprehensive coursera course, which really increase the utility of this work. I would therefore recommend publication.

I have just a few comments. Firstly, it would be helpful to present the separate impact of case isolation (D compartment) and quarantine of contacts (Q). Typically the latter will be smaller than the former, which could be achieved without TTI if symptomatic individuals self-isolated. This would help understand the added value of TTI over just isolation based on symptoms. Secondly, it is likely that willingness to be tested and self-isolate is related to the probability of being identified as a contact and being willing to quarantine (i.e. rho and omega in the detection matric are likely to correlated at the individual level). This will limit the impact of TTI and should be mentioned as a caveat. Finally, there is related, published work that uses a similar framework and comes to similar conclusions that ought to be mentioned (doi:10.1016/S1473-3099(20)30630-7).

Reviewer #3: Review of 'Maximizing and evaluating the impact of test-trace-isolate programs'.

In this manuscript, the authors develop a mathematical framework for analysing a test trace isolate (TTI) programme for COVID, aimed at helping public health authorities work out the weak spot of their programmes and improve them. There are two broad conclusions 1/ to be really effective, TTI can't have too many weak spots at all, and 2/ if the testing component is weak, fix that before looking at anything else. 

The mathematical framework is based on a next generation matrix formalism, and the outcome of interest is the effective reproduction number R. The paper is accompanied by an R package, an online page, and a Coursera online course; really useful work.

I think this is a very useful paper, very well put together. 

Here are some comments, hope they help improve an already excellent paper. 

You assume, but don't say in words, that self isolation and quarantine perfectly stop further transmission (can see in equations 5 and 6). That needs to be stated. I also wasn't too sure how to think about imperfectly effective isolation and quarantine in your framework. Please discuss (including in the Contessa website). 

The swap from calendar time to generation time is a neat trick in epidemiology (see e.g. Pellis Math Bioscience 2008), and used very effectively here. I was nonetheless left a bit puzzled by the incorporation of delays, and given the conclusions, I would have liked to understand this better. In short, I didn't see any detailed treatment of the delays beyond equations (5) and (6), which are only half defined, since f(x) and g(x) are not defined. I would have liked to see quite a bit more detail here to be satisfied that these results are reproducible. I would also have liked to see what effects the fact that tau_D and tau_Q are themselves distributed, and to have seen more information about the functions f(x) and g(x). We have found in our simulations that the variances of incubation, generation time, and delay times, all have quite a big effect. This was pointed out in Fraser et al PNAS 2004 cited here, and seems to be relevant for COVID.

There is a sensitivity analysis to different choices of generation time etc, but I would still nonetheless say that the baseline choice is a bit odd for COVID-19, consisting of a long generation time estimate (6.5 days, unreferenced?) compared to many publications (see e.g. Ganyani et al Eurosurveillance and many others), and effectively no asymptomatic infection. I wonder if the authors could rejig which are their baseline assumptions and which are sensitivity analyses.

Initially I was puzzled by the results which include focus on areas of parameter space where testing is poor, isolation is imperfect, and yet tracing and quarantine is highly effective. Yet, I can imagine this happening in practice due to these engaging very different expertise, and different public health departments. So in some ways it shouldn't need to be said that you then need to test more and faster, but it's good to set it out, and the findings here are clear.

Sup Table S3 does not say what value was used for the proportions of asymptomatic and symptomatic infections that are detected by surveillance, and I don't understand why. These two parameters are presumably different from each other and relevant for the results. 

The bottom half of Fig 2 should say quarantine instead of isolation. In Figs S1 and S2 the labels for the colours in the legend does not match the labels for the colours in the plot.

I haven't tried the R package. I think someone should before publication. It sounds like the Coursera materials have already been road tested, which is great. I tried the shiny app, very slick, though I wonder if some of the parameters are a bit obscure; e.g. the difference between being tested and being isolated. In our jurisdiction, people should isolate immediately after the test, a problem we face is that many don't. 

Signed - Christophe Fraser

[LINK]

---

## [Decision Letter · Decision Letter 2]

12 Mar 2021

Dear Dr Lessler, 

On behalf of my colleagues and the Academic Editor, Mirjam E. E. Kretzschmar I am pleased to inform you that we have agreed to publish your manuscript "Maximizing and evaluating the impact of test-trace-isolate programs: a modeling study" (PMEDICINE-D-20-04296R2) in PLOS Medicine.

PRESS

Open Science

Sincerely, 

Dr Raffaella Bosurgi 

Executive Editor 

PLOS Medicine